# Resilience Dynamic Assessment Based on Precursor Events: Application to Ship LNG Bunkering Operations

**Tomaso Vairo [1], Paola Gualeni [2], Andrea P. Reverberi [3] and Bruno Fabiano [1,*]**

[1] DICCA, Civil, Chemical and Environmental Engineering Department, Genoa University, Via Opera Pia 15, 16145 Genoa, Italy; tomaso.vairo@edu.unige.it
[2] DITEN, Electrical, Electronics and Telecommunication Engineering and Naval Architecture Department, Genoa University, Via Opera Pia 11A, 16145 Genoa, Italy; paola.gualeni@unige.it
[3] DCCI, Chemistry and Industrial Chemistry Department, Genoa University, Via Dodecaneso 31, 16146 Genoa, Italy; andrea.reverberi@unige.it
* Correspondence: brown@unige.it; Tel.: +39-010-3532585

**Abstract:** The focus of the present paper is the development of a resilience framework suitable to be applied in assessing the safety of ship LNG (Liquefied Natural Gas) bunkering process. Ship propulsion considering LNG as a possible fuel (with dual fuel marine engines installed on board) has favored important discussions about the LNG supply chain and delivery on board to the ship power plant. Within this context, a resilience methodological approach is outlined, including a case study application, to demonstrate its actual effectiveness. With specific reference to the operative steps for LNG bunkering operations in the maritime field, a dynamic model based on Bayesian inference and MCMC simulations can be built, involving the probability of operational perturbations, together with their updates based on the hard (failures) and soft (process variables deviations) evidence emerging during LNG bunkering operations. The approach developed in this work, based on advanced Markov Models and variational fitting algorithms, has proven to be a useful and flexible tool to study, analyze and verify how much the perturbations of systems and subsystems can be absorbed without leading to failure.

**Keywords:** Bayesian inference; decision support system; dynamic risk management; LNG ship propulsion; resilience engineering

## 1. Introduction

The new IMO Marpol Annex VI stringent requirements about ship exhaust gases have driven the use of innovative fuels and technologies that in turn, for some specific cases, might suggest the renewal of a safety paradigm. In this perspective, learning from high hazard industrial sectors offers novel opportunities, namely referring to resilience assessment. The last one integrates a set of key concepts to provide an innovative way of thinking about, and practicing, safety management. Resilience is fundamentally a system property. The application of this powerful concept is very versatile, and it represents an effective support to discuss safety performances of a complex system, i.e., when safety as a performance is the outcome of a successful interaction among different elements and sub-systems. This concept seems relevant in the case of a shore-to-ship LNG bunkering operation where the ship, the onshore infrastructure/asset and the connecting system are to be modelled and carefully analyzed in terms of interface and interference. In such complex cases, safety can be defined as an emergent property where resilience is the key enabling property. Moreover, the increasing interest about the resilience assessment is to be understood in the deep change of paradigm from the prescriptive approach to the performance-based one that continuous innovation in technology is asking for. As is widely acknowledged, risk assessment is a very useful approach in support of this change but at the same time it is not exhaustive to also capture the possible interface/interaction effects

among the several single components of a complex system besides the specific item-based failures. Risk analysis in a process plant is a matter of hazard identification and assessment of possible upset scenarios consisting of chains of cause-and-effect events modelled by fault and event tree, or bow tie. To this end, the Bayesian network has become the tool of preference, enabling overview, diagnosis of causes of disturbances, and predictive reasoning [1]. An extensive review on Bayesian networks and their applications in safety risk assessment and security risk assessment is provided by George and Renjith [2] and the reader is addressed to their paper for the most promising advancements within the process industries domain. Several contributions in recent years promoted dynamic risk assessment DRA methods. Basically, DRA is conceived as a methodology updating the estimated risk of a failing/deteriorating process according to the performance of the control system, safety barriers, inspection, and maintenance activities [3]. As originally suggested by Knegtering and Pasman [4], symptoms can be picked up by the signals or indicators that can warn of increasing risk factor strength influencing the reliability of process components and controls.

One of the reasons for the superior attention to resilience is the recent increased capability of data measurement/storage and relevant treatment for developing knowledge. The concept and the techniques to realize a resilient plant are still under investigation and are detailed in the seminal paper by Jain et al. [5] presenting the large potential with issues such as error-tolerant equipment design, receptive to early warning signals during operations, with a 'plasticity' response and effective emergency response. There is no single accepted set of components of resilience, so, the framework proposed in the present paper, which is strictly related to the state-of-art of scientific literature, represents a robust approach to a systemic vision of safety management. The above-described concepts are fully applicable to ship design and production, especially when dealing with very complex units like passenger ships. In addition, the need of a continuous innovation trend, together with the increasing importance of ship safety performances as emergent properties, constitutes a very interesting domain of application.

The applicative case study here investigated is relevant to a cruise ship, which represents a very challenging topic when attention is paid to environmental and accident hazards [6]. In fact, protecting sea environment and improving sustainable maritime traffic imply to extend the LNG use as fuel for ships other than LNG carriers, in addition to new technologies such as marine applications of fuel cells, possibly using green hydrogen to reduce greenhouse gas [7]. As an important aspect of the comprehensive and multifaceted cruise ship safety performance, the proposed study specifically focuses on the peculiar issue of LNG refueling activity. The introduced data-driven probabilistic methodology offers a means of predicting system resilience by analyzing the associated risks in terms of failure chances.

The remainder of this paper is structured as follows—Section 2 covers the resilience background and the conceptual framework addressing this study within LNG bunkering operation, followed by the methodology in Section 3. Section 4 presents an applicative case study to demonstrate the effectiveness of the proposed methodology, while the results and limitations of the study, followed by concluding remarks are outlined in Section 5.

## 2. Quantitative Risk Assessment (QRA) on LNG Bunkering Operation

The state of scientific literature on LNG bunkering Quantitative Risk Assessment is mainly related to the consequence analysis of LNG releases. Zhang [8] developed a conventional quantitative risk assessment method for determining the risk of LNG tanks operating close to ports. According to this study, the risk of LNG transportation was estimated to be within the acceptable risk range. Stokes et al. [9] focused on the need of QRA for novel ships using LNG as a fuel, suggesting that the human element constitutes the most important factor affecting safety of such an emerging design and operating requirements. The risk of LNG leakage during bunkering operations on pontoons was discussed in the work by Fan et al. [10], while Jeong et al. [11] introduced a quantitative

method for determining the safe exclusion zone for LNG bunkering operations. Following the recent trend towards inherent safety, Iannaccone et al. [12] assessed inherent safety of alternative bunkering technologies, based on consequence key performance indicators. Conventional fuel bunkering with fuel oil and marine gasoil are safer than LNG and therefore critical process units maybe identified and improved. A detailed risk matrix approach for LNG carriers approaching a bunkering terminal demonstrated its applicability, considering a case study in the port of Venice [13]. From a literature survey, the resilience of the LNG bunkering operation and its application into a dynamic risk assessment framework represents a research item still unexplored and well worth investigation. Additionally, in the traditional view of safety as a non-event, there is a difficulty to define the domain of attention of where to focus the effort and manage the performance. In this regard, resilience is considered an important capability needed by the 21st century systems [14].

In the latest decades, the term resilience has overflown from the material science (ability of a material to absorb energy when it is deformed elastically and release that energy upon unloading) to other different fields like ecology, psychology, infrastructures and complex systems in general. It should be remarked that environmental risk assessment within the wide framework of the Seveso Directive is an appealing research area, still under development, and bringing out novel topics to be thoroughly discussed and faced by advanced tools [15]. Resilience has been defined in the literature as "the ability of the systems to adapt to changing conditions in order to maintain a system property" [16], for example, safety. In other words, "a system is resilient if it can adjust its functioning prior to, during, or following events (changes, disturbances, and opportunities), and thereby sustain required operations under both expected and unexpected conditions" [17]. The latter definition seems more practically oriented and "sustain required operations" can be intended as the capability to safely carry out an operation. Summarizing, an ideal resilience analysis should hedge us for the unexpected and unknown in the given domain. In line with the observations by Pasman et al., [18] utilizing Big Data and Analytics hype to derive information on various kinds of observables, including weak signals, enables predictive assessments of equipment reliability/safe life, and provides lagging and leading indicators, including resilience ones. However, even though resilience seems to be a very promising support, quantitative metrics of resilience are not well established and further investigation about approaches and techniques is needed [19,20]. In the following, we outline a data analytic-based methodology and present its application in the context of an LNG refueling operation for a ship. Hollnagel [21] proposed four capabilities for resilient performances, namely the ability to respond, the ability to monitor, the ability to learn and the ability to anticipate. As amply reported, the quantitative risk assessment (QRA) process is crucial for an effective control of major accident hazards even if it is affected by several limitations, essentially connected to its inherent static nature. Newly developed frameworks, including dynamic ones, were recently developed, and applied also to improve the effectiveness of accident investigations, e.g., [22]. A main weakness is also represented by the large error bands associated with data for the likelihood associated with equipment outcomes, e.g., the likelihood of leaks of different size spills from pipes, valves, obtained from various published sources. The focal point to assess the resilience of a system relies in the identification of precursor events, which refers to early detection of "weak" signals from the system during the operations [23]. To identify the precursor events and thus maintain stability by applying appropriate adjustments, the analysis of a large amount of data is needed. By the data analysis, it is possible to predict the behavior of the system, thus catching the resilient performance according to the above mentioned four guidewords. The importance of the approach is remarkable also in different contexts, e.g., anticipation, absorption through robustness and redundancy, adaptation and recovery are the key attributes recently adopted in developing framework for enhancing the resilience of critical infrastructure to climate change [24]. During the last decade, the so-called data-driven models have increased their development and application. These models rely upon the methods of computational intelligence and machine learning and thus assume

the presence of a considerable amount of data, able to describe the modelled system's physics. As commented by Sarkar et al. [25], accidents do not occur in a chaotic fashion, so underlying patterns and trends do exist and can be captured. More recently, starting from the classification in domains explaining the complexity of the human–machine interaction, a novel approach was developed accounting for the information processing of the human brain [26]. In the field of safety-critical industries, a risk assessment approach based on machine learning developing a deep neural network model was successfully tested considering a drive-off scenario involving an oil and gas-drilling rig [27]. Data-driven modelling can also be considered as an appropriate approach to resilience assessment that would complement the "knowledge-driven" models describing physical behavior. Jianbin et al. [28] proposed, for stochastic non-linear systems, an input delay method for describing the sample-and-hold behavior of outputs, which is applied in a Fuzzy-affine model. The Bayesian approach has been proven to be a robust probability reasoning method under uncertainty, providing a tool for incorporating evidence during operations. State-of-the-art application of BN in FTA for systems for which the minimal link sets (MLSs) and minimal cut sets (MCSs) are known was presented in [29]. Different approaches in the field of Bayesian reasoning have been proposed for modelling a system under stochastic conditions, and the use of Hidden Markov Models (HMMs) seems to be one of the most promising and reliable [30]. HMMs allow for performing a forward and backward inference, can be used to conduct operational reliability analysis in complex systems [31,32], and will be adopted as reference tool, as detailed in the following.

## 3. Theoretical Framework

Starting from the operational steps of the LNG bunker activity in the maritime field, various coupled BNs can be built, which involve the probability of operational perturbations together with their updates, based on the hard (failures) and soft (process variable deviations) evidence during the operation. Ship propulsion by LNG as a possible fuel (with dual fuel engines installed on board) is becoming a more and more favored option, especially in the cruise ships market. However, this innovative solution implies the need to deepen some safety issues that might be involved in the LNG bunkering operations. When dealing with flammable HazMat, the potential loss of containment must be considered of primary importance in relation to storage tanks and piping, where in the case of an accident the dominant scenario is pool fire [33]. However, the probability of a scenario evolution can be affected by large uncertainties in its evaluation, for example, connected to the possibility of immediate, or delayed ignition [34]. Additionally, potential complex inter-dependencies between risk factors and reliance on deterministic probability values can add further uncertainty to the actual risk. Not so many investigations are available in the literature at present and this study aims to frame the most significant critical aspects of such probability evaluation.

The logic diagram for the proposed resilience assessment framework, in terms of stepwise procedure, is depicted in Figure 1, where the previously recalled capabilities (monitor, learn, anticipate, and respond) identifying resilience performances can be pointed out.

Starting from these premises, the integrated approach to carry out a resilience assessment in complex systems is summarized as follows.

### 3.1. Identification of Weak Signals

A weak signal indicates a possible degradation of the system's resilience and represents a decreased ability to cope with unexpected and unforeseen disruptions. They are seemingly random or disconnected pieces of information that at first appear to be irrelevant but can be recognized as part of a significant pattern by viewing them through a different frame or connecting them with other pieces of information. Weak signals can be identified starting from the risk assessment process and can be found in the Fault Trees MCSs (Minimal Cut Sets), by analyzing how the process variables oscillate around the set points.

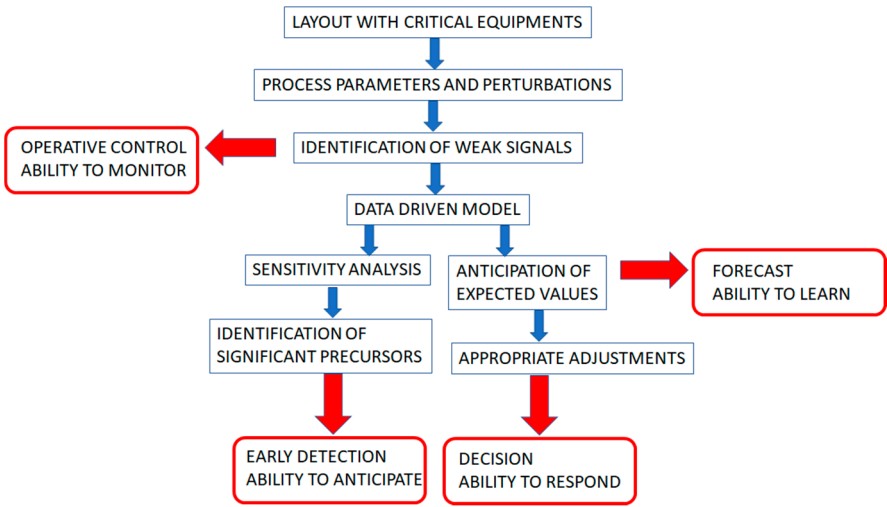

**Figure 1.** Resilience assessment framework.

Monitoring the critical process variables and their deviation from the set points allows establishing appropriate operational control strategies.

### 3.2. Data Driven Models and Precursors Identification

As widely discussed, the focal point of ML-based models is the investigation of data. All the dependencies, correlations, inference statistics can be found in data, so it is crucial to build a good data-driven model for extracting all the information usually contained in the data. It is thus possible to identify the significant perturbations and, by training the model, anticipate the systems outcome, to improve decision-making and promptly choose the appropriate adjustments.

As previously anticipated, for analyzing the significant parameters perturbations, and thus identifying and anticipating the weak signals, a Hidden Markov Model (HMM) has been developed. An HMM is a generative probabilistic model, in which a sequence of observable X variables is generated by a sequence of internal hidden states Z. The hidden states are not observed directly. The transitions between hidden states are assumed to have the form of a first order Markov chain. They can be specified by the start probability vector $\pi$ and a transition probability matrix A. The emission probability of an observable can be any distribution with parameters $\theta$ conditioned on the current hidden state. The HMM is completely determined by $\pi$, A and $\theta$.

In the present study, the hidden states are the states between a regular performance and a failure of a component. The only known states are the first (the component is performing well) and the last (the component fails), and the hidden states in between may represent the precursors of accidental events. The emissions of the system are the process variable values.

Three possible implementations of the HMM are evaluated.

- In the first implementation, besides the observations, also the transition probabilities (derived a priori from FT, for the last state), and the emission probabilities (derived from expert knowledge) are inserted, in the form of a transition matrix and emission matrix. The model determines the most likely sequence of states by inference (MC sampling with rules) on the observations;
- The second model has the same observations and transition probabilities as the first one. The emission probability and the most probable sequence of states are determined by inference; and
- In the third model, only the observations are given. There is no information on either transition or emission probabilities. The model can infer all the information and determine the most likely sequence of states.



The HMMs are developed in python using the packages PyMC3 and Theano [35].

PyMC3 has been used for the implementation of the Metropolis–Hastings (MCMC-MH) algorithm to perform forward and backward inference by computing the distribution space of the model parameters and determine the most likely outcome. This technique requires a simple distribution called the proposal Q $(\theta' \mid \theta)$ to help draw samples from an intractable posterior distribution P $(\theta = \theta \mid D)$. These two distributions are called conjugate distributions.

MH uses Q to randomly walk in the distribution space, accepting or rejecting jumps to new positions based on how likely the sample is.

To decide if $\theta'$ is to be accepted or rejected, the following ratio must be computed for each new proposed $\theta'$:

$$\frac{\prod_i^n f(d_i|\theta = \vartheta\prime)P(\vartheta\prime)}{\prod_i^n f(d_i|\theta = \vartheta)P(\vartheta)} \tag{1}$$

where f is the above-mentioned proportional function. The acceptance rule is set as follows:

If (Equation (1)) < 1: P(accept) = (Equation (1))

If (Equation (1)) $\geq$ 1: P(accept) = 1

This means that if a $\theta'$ is more likely than the current $\theta$, then $\theta'$ will always be accepted. If it is less likely than the current $\theta$, then it might be accepted or rejected randomly with decreasing probability, the less likely it is.

By varying one input at a time (process variables values) and correspondingly analyzing the output sequences, it is possible to perform a sensitivity analysis to determine the most critical variable.

## 4. Applicative Case Study

At present, the use of LNG as fuel is among the most successful solutions to comply with the MARPOL Annex VI requirements. This is in fact the most popular solution at present for new buildings, especially in the field of cruise ships. Solutions adopt a dual fuel engine able to use both oil and gas—according to this configuration, scrubber systems and low-sulfur fuel are not required. As is amply known, LNG releases no sulfur, 99% less particulate emissions, 85% less NOx emissions, and 25% less greenhouse gas emissions. Consequently, LNG can be regarded as an inherent cleaner fuel allowing to obtain a sharp reduction of critical pollutant emissions [36]. The most notable benefits and advantages of using LNG are cleaner emissions and lower cost. The LNG-based fuel system technologies show a better sustainability performance than the conventional marine fuel technologies [37]. LNG is natural gas cooled to approximately –260 °F ($-162.7$ °C) and it is reasonably easy to store and transport it. In its liquefied state, LNG is odorless, colorless, non-toxic, and non-corrosive.

A main issue connected to the adoption of LNG as fuel is the lack of bunkering (fueling) facilities available yet, so getting an LNG-powered ship re-fueled may be problematic. Though there are plans for more fueling depots to be established to serve LNG-fueled ships, some worries are related to the possible threat that this plant and its operational activity might create—several administration and port authorities are addressing the safety issues the refueling operations might create for citizens and coastal environments. By the way, the item of ship-loading of hydrocarbons and the assessment of the health risk associated with these sources of contamination is not widespread as it is currently not addressed by the European Directive on the integrated pollution prevention, nor by other environmental regulations [38].

Basically, the possible technical solutions under the current development in Europe and worldwide are [39]:

- Truck-to-Ship—TTS;
- Ship-to-Ship—STS; and
- Terminal (Port)-to-Ship—PTS

For the purpose of this paper, the resilience assessment will be carried out for the case of shore-to-ship refueling and a schematic layout is depicted in Figure 2 [40]:

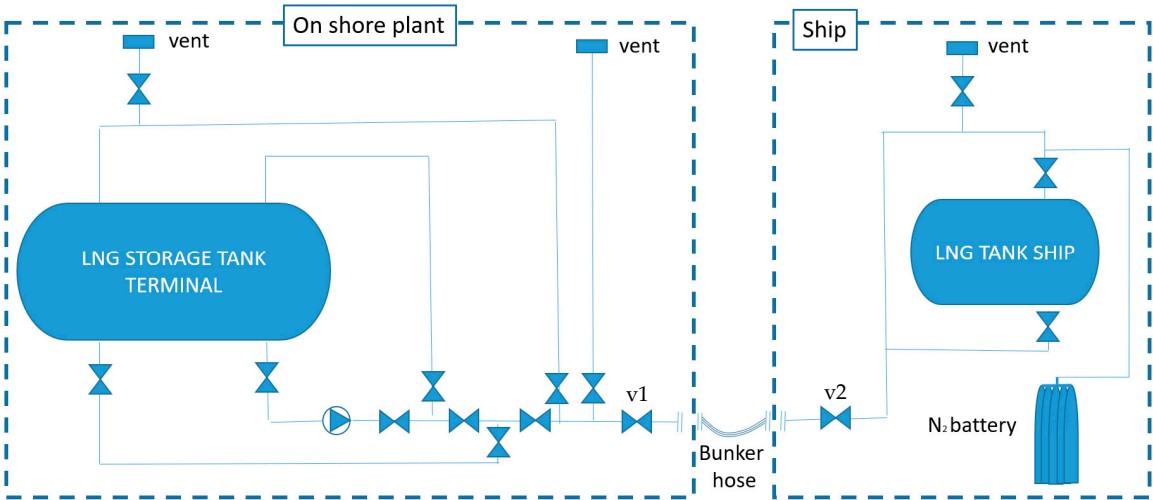

**Figure 2.** Representative layout of a shore-to-ship LNG refueling plant.

The basic steps of the bunkering process are summarized as follows, derived with some simplification from EMSA:

- Precooling of the line (landside), cargo pump included;
- Actions to avoid ground fault arcing;
- Loading arms are usually used for bunker hose connection;
- The hose is put in place;
- Inert gas is used to remove oxygen and moisture from the piping of the receiving ship;
- Then, the receiving system is purged from the residual nitrogen using the natural gas remained in the LNG tank on board the ship;
- Closure of the onshore side valve (v1);
- Closure of the ship side valve (v2);
- Liquid line stripping;
- Bunker line inerting; and
- Disconnection of the bunkering hose.

For the purpose of this applicative study, specific attention will be given to the liquid transfer phase, with focus on the following critical steps:

- Analysis during the actual bunkering phase; and
- Analysis during the immediate post-bunkering phase with the pressure increment.

As already discussed, resilience analysis requires in the first instance to make a distinction between the condition at a given moment of time, i.e., static (performed adopting conventional QRA approaches), and trends during plant operation, i.e., dynamic (relying on innovative tools). The whole resilience assessment can be very large; to understand the validity of the proposed methodology the investigation will be limited to the leakage hazard originating in the part of the system between the two flanges of the connecting hose, technically indicated as "LNG transfer system" [41].

*4.1. Fault Tree Analysis*

The FTA for the bunkering operations is carried out starting from technical reports [42]. The following Tables 1 and 2 summarize the FTA elements.

**Table 1.** Equipment Count for Leak Frequency Estimate.

| Root Component | Quantity (Diameter) |
|---|---|
| Manual valves | 3 (3 in.) |
| Activated valves (ESDs) | 2 (3 in.) |
| Flanges | 12 (3 in.) |
| Small bore fittings | 2 (1 in.) |
| Flexible hose | 1 (3 in.) |
| Manifold piping | 100 m (3 in.) |

**Table 2.** Risk assessment assumptions.

| Client Type | Source (m$^3$) | Client (m$^3$) | Rate (m$^3$/h) | Op. time (h) | Freq (occ/y) |
|---|---|---|---|---|---|
| Ferry | | 200 | 50 | 4 | 365 |
| OSVs | 500 | 400 | 200 | 2 | 183 |
| Container | | 2400 | 600 | 4 | 52 |

The developed FTA related to leakage hazard originated in the LNG transfer system is shown in Figure 3.

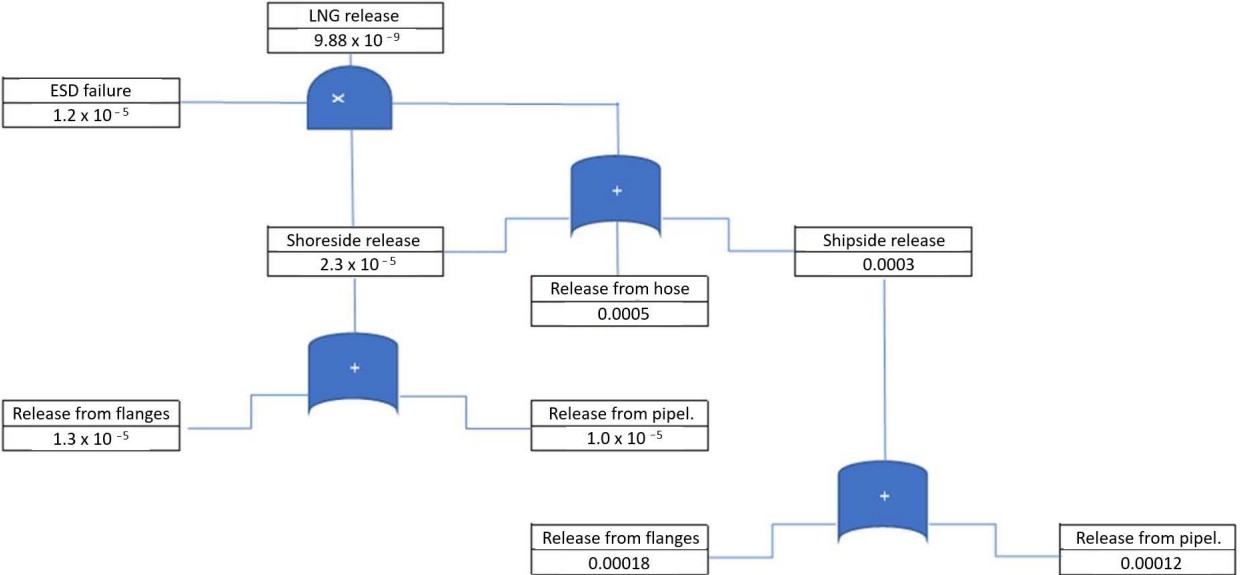

**Figure 3.** Fault tree developed for LNG loss of containment (LOC).

The failure probability of the single component are taken from the literature [43].

The relevant process variables in the LNG bunkering operations are pressure, temperature, and bunkering rate. The conventional pressure in a bunker hose is around 5–6 bar (g) and the corresponding LNG temperature during bunkering is around −145 °C, based on the assumption that bunkering operations can maintain a constant temperature by managing the boil-off vapors.

### 4.2. The Bayesian Perspective on Risk Assessment

A conventional fault tree has a converging structure that describes how a group of root events can lead to a top event. This logical structure enables causality reasoning between root events and a top event; it allows performing both forward and backward analysis. For quantitative reasoning, however, only statistical and static information is available. To calculate the probability of the occurrence of a top event, the probabilities of the root events have to be either estimated from statistical data or specified by expert knowledge. Furthermore, the basic events are assumed to be statistically independent [44].

As previously discussed, the limitations of FTAs can be overcome by using the Bayesian probabilistic approach and applying the MCMC-MH algorithm to update the basic event on real time information, thus reducing uncertainty and capturing the dynamic behavior of the system.

The FTAs can be dynamically updated by considering the root failures frequency as prior probabilities, and then performing a double-trace MCMC-MH (continuous and dotted traces) simulation (only 5% of original population—evidence—is observed). The posterior probabilities are estimated with reference to the solution of the FTAs, as depicted in Figure 4.

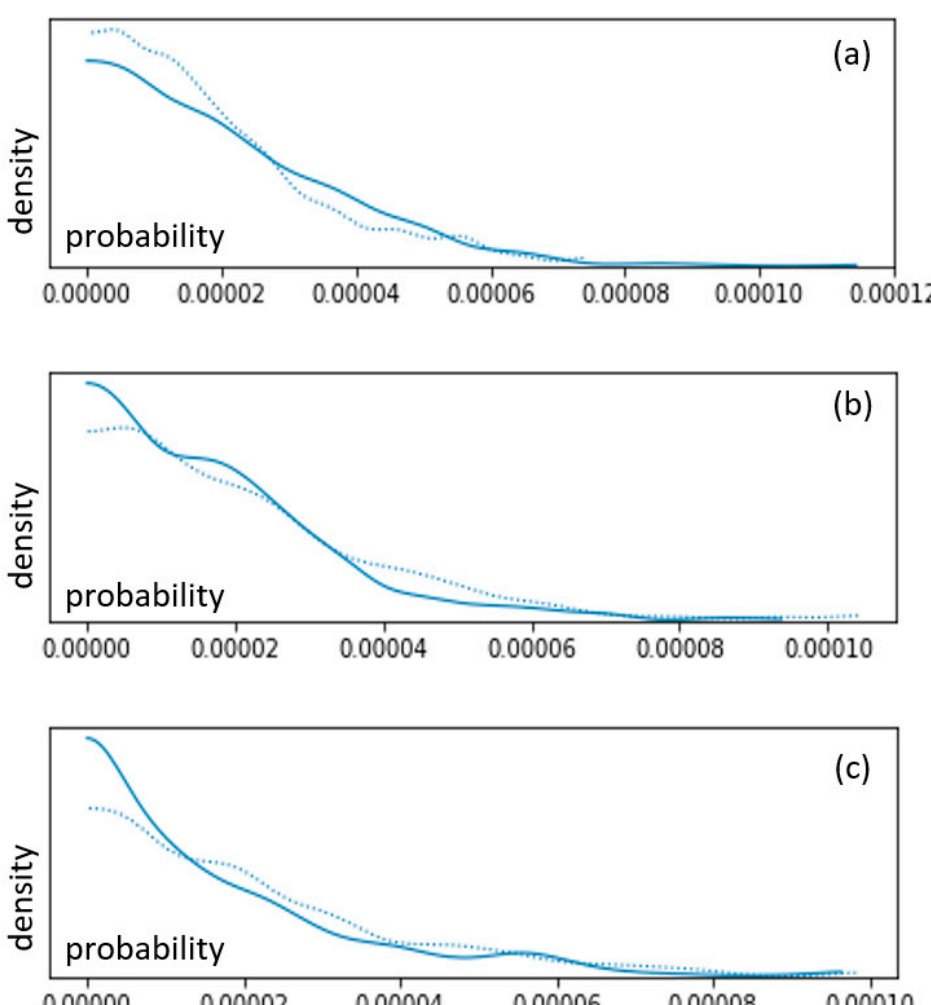

**Figure 4.** PDF of leakage on the shoreside plant (**a**), shipside plant (**b**) and connection hose (**c**).

The posterior probabilities are then evaluated by the means of HPD (highest posterior density), as shown in Figure 5.

At last, by combining the contributions, it is possible to obtain the posterior predictive PDF as exemplified in Figure 6, where the red line indicates the mean expected value.

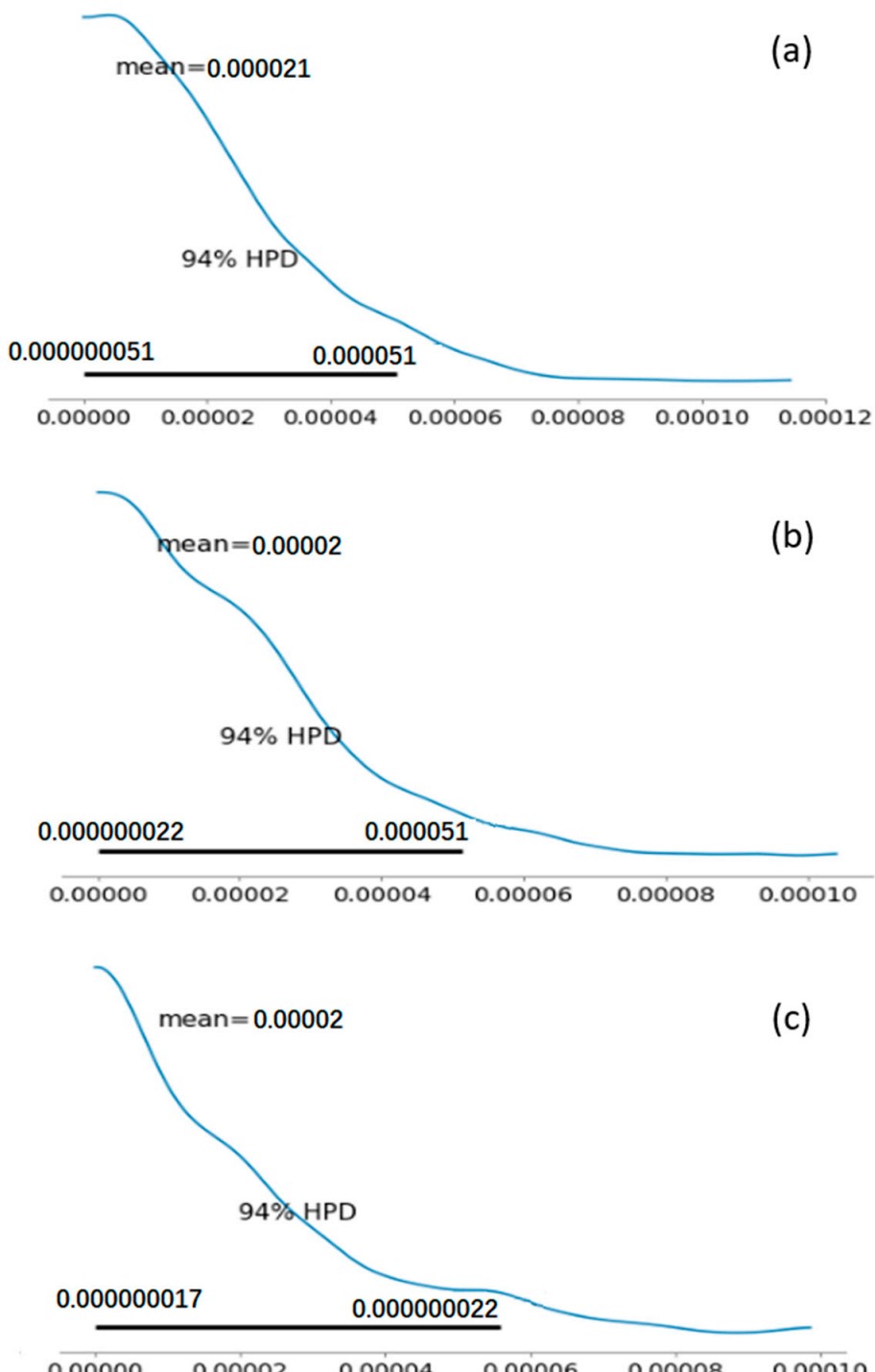

**Figure 5.** Posterior probabilities distribution for the leakage on the shipside (**a**), shoreside (**b**) and connection (**c**).

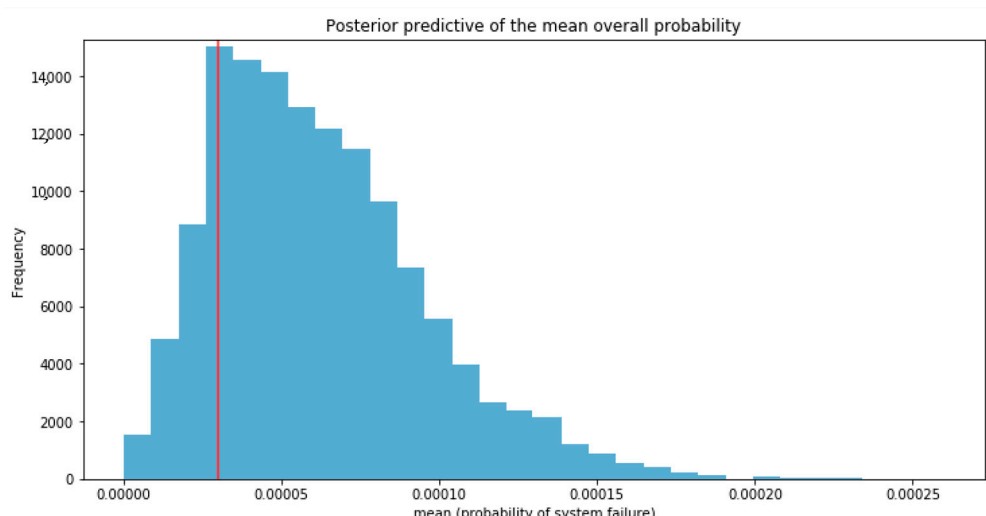

**Figure 6.** Posterior predictive distribution and mean probability of system failure.

### 4.3. State Sequence Prediction

Starting from the basic dynamic approach outlined, a basic step with respect to maintaining an adequate safety level is connecting the failure probabilities with the system states. All the detected signals and indicators will be properly treated by the HMM approach, as described in the following. The input parameters are the process variables, pressure, temperature and density, each one with appropriate set points as detailed in the following.

LNG line process equipment and hose

- Operating pressure is set to 10 bar(g). This is the maximum operating pressure for LNG process equipment according to European design standard EN1472-2;
- Operating temperature is set to $-162$ °C to keep the inventory in liquefied state. The bunker vessel (discharging unit) is assumed to be able to maintain this constant temperature during the transportation to site; and
- Density depends on temperature and pressure. Based on the defined process parameters the density is 425 kg/m$^3$

Vapor return line (NG)—process equipment and hose

- Pressure is set to 2 bar(g) as it will be reduced compared to LNG line;
- Temperature is set to $-100$ °C. The liquid has been warmed and is now in a vapor state; and
- Density 4.3 kg/m$^3$.

Tank

- The pressure in the tanks is set at 2 bar(g).

The operational parameters are acquired by the model using json (Javascript object notation), which is a data interchange format, and it is directly usable in python [35].

The input dataset for each sub-section is provided in Table 3.

**Table 3.** Head of an input dataset.

| Timestamp | Pressure (Barg) | Temperature (°C) |
|---|---|---|
| 2020-10-21 10.10.00.000 | 9.88 | $-162.02$ |
| 2020-10-21 10.15.00.000 | 9.91 | $-162.04$ |
| 2020-10-21 10.20.00.000 | 9.99 | $-162.03$ |

An example of the HMM sampling is shown in Figure 7, obtained by considering the intermediate hidden state. In the same Figure 7, s0-sn represent the simulated future time-steps, while the bar-charts on the left hand side represent the most probable expectations deriving from the inferential sampling shown on the right hand side (nr. of MCMC samples vs. state).

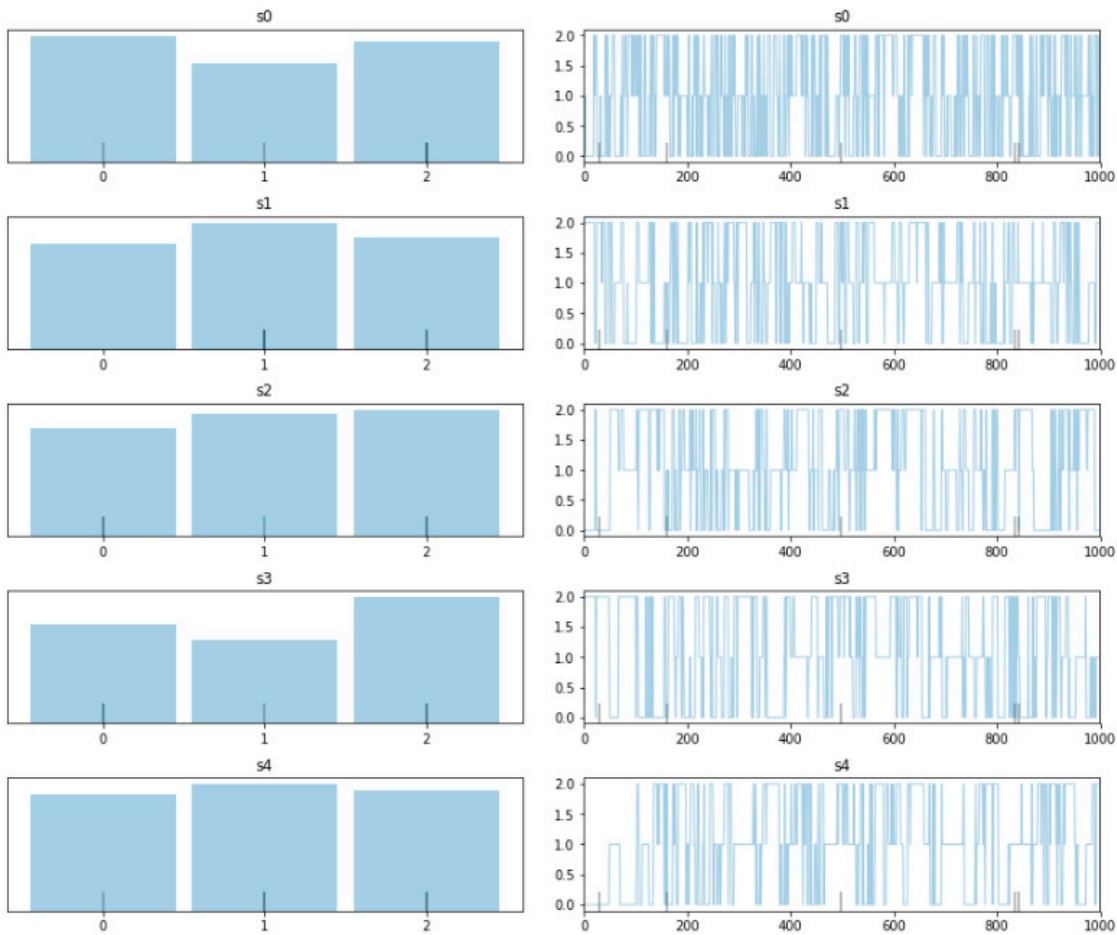

**Figure 7.** Excerpt obtained by HMM inferential sampling and prediction.

By combining all the most probable outputs, the most probable state sequence of the system is at last obtained in form of immediate readability, as depicted in Figure 8.

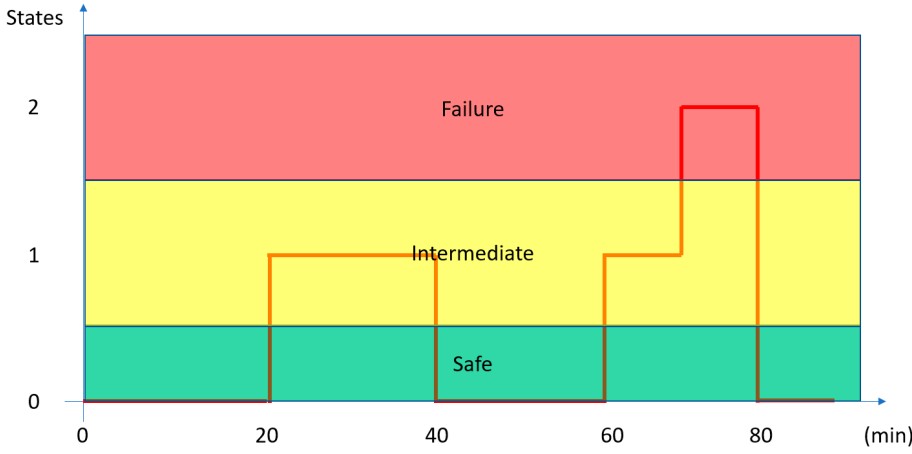

**Figure 8.** States sequence prediction and resulting resilience indicator.

### 4.4. Key Resilience Considerations

From the simulations described above, root failures (and consequently leakages) can be anticipated by analyzing the data coming from the operations. According to the outlined approach, the failure state is anticipated by the intermediate state, allowing an early warning. By analyzing the HMM traces, it is possible to determine the probability for each state of the given system. Additionally, based on the model outputs making backward inference, one can immediately refer the hazardous deviation to some sub-systems failure. This can strengthen the understanding and application of early process fault detection in effective process risk management.

Table 4 summarizes the expected probabilities for the states of the root components (expressed in terms of MAP—Maximum A Posteriori), obtained by the Resilience model and compared with the results of the traditional FTA.

**Table 4.** Expected probabilities of occurrences of possible states in the root components.

| Root Component | Traditional FTA | Resilience Model |
|---|---|---|
| SHORESIDE VALVES | | |
| Safe | 0.999 | 0.228 (MAP) |
| Intermediate | NA | 0.761 (MAP) |
| Fail | $1.2 \times 10^{-6}$ | $1.6 \times 10^{-8}$–$1.6 \times 10^{-5}$ (94%HPD) |
| PUMP | | |
| Safe | 0.999 | 0.166 (MAP) |
| Intermediate | NA | 0.833 (MAP) |
| Fail | $1.3 \times 10^{-6}$ | $1.8 \times 10^{-8}$–$1.8 \times 10^{-5}$ (94%HPD) |
| SHORESIDE PIPELINE | | |
| Safe | 0.999 | 0.387 (MAP) |
| Intermediate | NA | 0.612 (MAP) |
| Fail | $1 \times 10^{-6}$ | $1.7 \times 10^{-8}$–$1.7 \times 10^{-5}$ (94%HPD) |
| HOSE | | |
| Safe | 0.999 | 0.055 (MAP) |
| Intermediate | NA | 0.854 (MAP) |
| Fail | $7.5 \times 10^{-6}$ | $5.7 \times 10^{-9}$–$1.8 \times 10^{-5}$ (94%HPD) |
| SHIPSIDE VALVES | | |
| Safe | 0.999 | 0.297 (MAP) |
| Intermediate | NA | 0.702 (MAP) |
| Fail | $1.8 \times 10^{-6}$ | $8.7 \times 10^{-9}$–$1.7 \times 10^{-5}$ (94%HPD) |
| SHIPSIDE PIPELINE | | |
| Safe | 0.999 | 0.307 (MAP) |
| Intermediate | NA | 0.692 (MAP) |
| Fail | $1.2 \times 10^{-6}$ | $1.2 \times 10^{-8}$–$1.7 \times 10^{-5}$ (94%HPD) |

Recalling the four needs for resilient performance, the following points can be summarized:

- The model allows identifying how the state of the plant is changing over time, thus detecting the occurrences of perturbations during the operations and responding to the perturbation. The intermediate state defines the precursor of a perturbative event;
- The approach is able to monitor by analyzing in real time the data derived from the plant, finding the corresponding actual state;
- Through the learning Bayes-based algorithm, the model can produce increasingly reliable forecasts on the progress of the operation, as the training dataset is constantly updated by the actual operative evidence; and
- By identifying the precursor events, the model anticipates the states transitions, providing an early warning to take appropriate countermeasures.

The probabilistic nature of the perturbations, and thus of their associated outcomes, requires a probabilistic scoring system for resilience. Additionally, the multitude of conceivable scenarios, each one with associated probability distribution, necessarily limits any scoring system to specific classes of representative perturbation, without prejudice to the

possibility of inserting new ones derived from the system application in the field. The resilience of the whole system can thus be expressed as an indication of how likely the system is changing its state. Combining the most probable sequence of system states (Figure 8), with a Monte Carlo asynchronous sampling from the posterior predictive probability of failures (Figure 6), into the same HMM model, it is possible to represent the resilience score with a single parameter R varying between 1 (corresponding to safe system mode) and 0 (corresponding to failure system mode). The value of R can be continuously updated with the state evidence obtained by the previously mentioned random walks. Each step of the overall resilience score considers the CDF (Cumulative Distribution Function) of the different state probability.

Figure 9 represents how the resilience score R is changing during the operation, when different perturbative situations appear. In this way, the approach would open the way for continuous monitoring of the resilience level and provide anticipated indications for when and where to adopt corrective actions. The last item is well connected to the correct implementation of effective planning and execution of emergency response, which is recognized as a key learning lesson from accidents in the process industries [45].

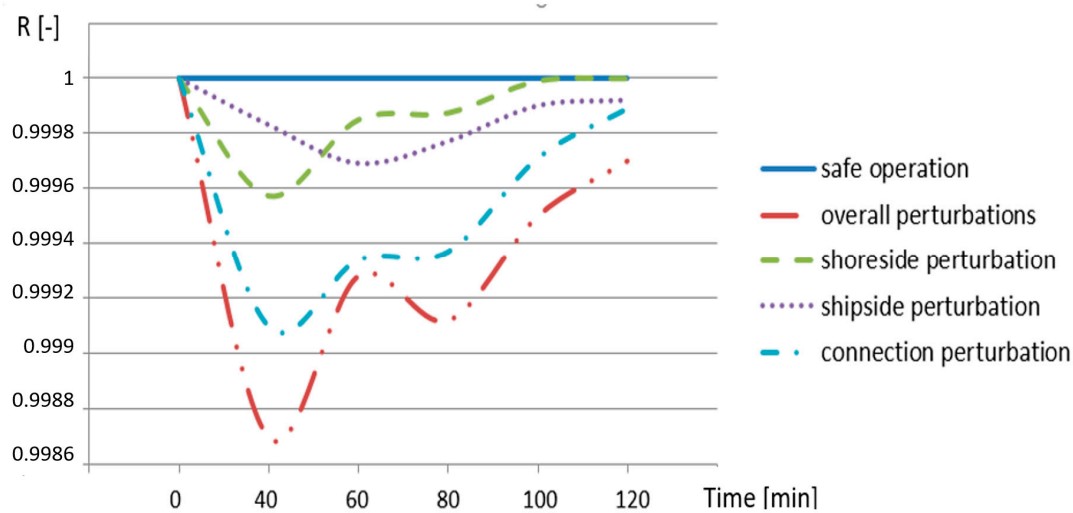

**Figure 9.** Value of the resilience score over time, corresponding to different perturbative situations.

## 5. Conclusions

The idea behind the resilience analysis is that safety is an emerging property of the system that can be framed with the four guidewords, i.e., monitor, learn, anticipate and respond. This work has shown how it is possible to evaluate the system's resilience through dynamic analysis, connected with what is happening in the plant at that precise moment in the operation in progress.

The main features of the model are the following:

- A dynamic representation of the loss of containment risk, related to the values of the process variables, is obtained by combining a Bayesian network for inferential sampling, with an HHM in a resilience model for the determination of hidden states probabilities; and
- The sequences of the most probable system states represent relevant information for taking the most appropriate actions on time, in order to avoid potentially hazardous situations.

The approach used in this work, based on Bayesian statistical modelling and probabilistic machine learning, which focuses on advanced Markov Models and variational fitting algorithms, has proven to be a useful and flexible tool to study, analyze and verify the achievement of the four basic needs of the resilience paradigm. The proposed resilience

score R can represent a valid metric to define how much the perturbations of systems and subsystems can be absorbed without leading to failure. Furthermore, performing a resilience assessment can help decision-makers and planners to pursue environmental and safety objectives more effectively. As far as the specific application case, in relation to the plausibility of assumed input data, it seems that the LNG bunkering operation of a ship is characterized by satisfactory resilience properties. Results can show that the risk due to the probability of leakage is low. Even though this is only one of the possible hazards worth investigating with this approach, this quantitative evidence on the dynamic perturbation probabilities can be further validated and used in a rational decision-making process about new LNG refueling plant installation. As a main limitation of the study, it must be said that by monitoring resilience over time, only an indication could be given about the circumstances in which the fluctuation of perturbations might lead to a loss of containment. Preliminary results show that they are characterized by reduced probability intervals, with values by far below the acceptability threshold. It is believed that upon proper refinement the approach can be effectively used to capture the dynamic evolution of internal and external risk conditions and effectively support critical decisions to improve the overall safety of the workers. This work represents a first attempt, requiring further experience and validation, bearing in mind that as already mentioned, a comprehensive resilience analysis should include all representative hazard indicators, supported by field data. Further development of this potentially versatile and robust approach could entail the environmental parameters as well, investigating their influence on the resilience assessment.

**Author Contributions:** Conceptualization, T.V. and B.F.; methodology, T.V.; validation, P.G. and A.P.R.; data curation, T.V., P.G.; writing—original draft preparation. T.V. writing—review and editing, T.V., A.P.R.; supervision, B.F., P.G. All authors have read and agreed to the published version of the manuscript.

**Funding:** This research received no external funding.

**Informed Consent Statement:** Not applicable.

**Data Availability Statement:** Not applicable.

**Conflicts of Interest:** The authors declare no conflict of interest.

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
