# Peer review of "Resilience Dynamic Assessment Based on Precursor Events: Application to Ship LNG Bunkering Operations"

_sustainability, doi:10.3390/su13126836_

Round 1

Reviewer 1 Report

This study developed a resilience framework to be applied in assessing the safety of ship LNG bunkering process. Overall, the paper is well-structured, the methodology is presented explicitly, and the case study is thoroughly discussed. There are just some minor issues that need to be addressed:

  1. Abstract: more content of the major results or findings of this study should be added in the abstract.
  2. Typo: Line84: "2. 2. QRA on LNG bunkering operation" should be "2. "
  3. Line 129: "quantitative risk assessment (QRA)" should be defined when it showed up the first time, such as line84 or line89.
  4. Figure 4,5,6 should be presented in one figure with panel numbers of (a)(b)(c), respectively. Labels of the x-axis and y-axis need to be added. 
  5. Figure 7,8,9 should be modified the same way as point 3 mentioned.
  6. Table 3: add the unit of Pressure and Temperature. 
  7. Figure 11: add labels for the x-axis and y-axis.

Author Response

Reviewer 1

We much appreciate comments of reviewers and their suggestions that certainly improve the overall appeal of the manuscript. We believe that this revised version of our manuscript incorporated the suggested changes and points the reviewers mentioned. 

Abstract: more content of the major results or findings of this study should be added in the abstract.

Response: The abstract is extended with a flavour of main results and conclusion.

Typo: Line84: "2. 2. QRA on LNG bunkering operation" should be "2. "

Response: The typo is fixed together with other minor typos throughout the text.

Line 129: "quantitative risk assessment (QRA)" should be defined when it showed up the first time, such as line84 or line89.

Response:

As correctly suggested, the definition is inserted in the chapter tittle.

 Figure 4,5,6 should be presented in one figure with panel numbers of (a)(b)(c), respectively. Labels of the x-axis and y-axis need to be added. 

Response: Figures 4, 5 and 6 are integrated into Figure 4.

 Figure 7,8,9 should be modified the same way as point 3 mentioned.

Response: Figures 7, 8 and 9 are integrated into Figure 5.

Table 3: add the unit of Pressure and Temperature. 

Response: The units are added.

Figure 11: add labels for the x-axis and y-axis.

Response: The figure depict the MCMC sequence, and the explanation of the x and y axis are added in the text.

English style of the whole manuscript was revsied for typos and improved.

Reviewer 2 Report

Thank you for the nice application on LNG bunkering. The reviewer comments are as follows:

  1. Typo on page 2 line 61, "ad" should be "and"
  2.  On Section 3, what do you mean by hard and soft evidence? please clarify it.
  3. Number and label the most relevant valves in Fig. 2 for the sake of clarity.
  4. In Page 7, the step of the bunkering process (lines 296 to 308), name the involved valves in Fig. 2 for the sake of clarity.
  5. Explain the meaning of dotted and solid lines in Figs. 4, 5, and 6.
  6. What is the y-axis in Figs. 7 to 9? Also, avoid using the notation **E-2, use instead **x10-5.
  7. Add the reference for using json (Page 11, line 395).
  8. In Fig. 10, the distribution is not normal,  it is skewed to low mean(probability) which the reviewer thinks is good. However, the reviewer wonders if other statistical information (mode, mean, median, and standard deviation?) will be worthy or enrich the discussion. Does the red line in the figure mean mode? please clarify it.

Author Response

Thank you for the nice application on LNG bunkering.

Many thanks for your kind note and constructive feedback. We have reflected on your comments and suggestions and revised the manuscript accordingly.

Typo on page 2 line 61, "ad" should be "and"

Response: The typo is fixed.

On Section 3, what do you mean by hard and soft evidence? please clarify it.

Response: The complete explanation is added. many thanks for the constructive advice.

Number and label the most relevant valves in Fig. 2 for the sake of clarity.

Response: The indication of the two critical valves is inserted.

In Page 7, the step of the bunkering process (lines 296 to 308), name the involved valves in Fig. 2 for the sake of clarity.

Response: The indication of the two critical valves is inserted.

Explain the meaning of dotted and solid lines in Figs. 4, 5, and 6.

Response: The meaning of the two traces is inserted.

What is the y-axis in Figs. 7 to 9? Also, avoid using the notation **E-2, use instead **x10-5.

Response: On the left side are barcharts of the most probable outcomes, so the x axis is the state and the y axis is the nr. Of outcomes. On the right side are the MCMC sampling, so the x-axis is the nr. Of samples and the y-axis is the state. The explanation is added.

Add the reference for using json (Page 11, line 395).

Response: The correct reference is added, many thanks for the right suggestion.

In Fig. 10, the distribution is not normal, it is skewed to low mean(probability) which the reviewer thinks is good. However, the reviewer wonders if other statistical information (mode, mean, median, and standard deviation?) will be worthy or enrich the discussion. Does the red line in the figure mean mode? please clarify it.

Response: The red line is the expected value, i.e. the most probable prediction. Reference is added in the text
